# UNDERSTANDING RARE SPURIOUS CORRELATIONS IN NEURAL NETWORKS

## ABSTRACT

Neural networks are known to use spurious correlations such as background information for classification. While prior work has looked at spurious correlations that are widespread in the training data, in this work, we investigate how sensitive neural networks are to *rare* spurious correlations, which may be harder to detect and correct, and may lead to privacy leaks. We introduce spurious patterns correlated with a fixed class to a few training examples and find that it takes only a handful of such examples for the network to learn the correlation. Furthermore, these rare spurious correlations also impact accuracy and privacy. We empirically and theoretically analyze different factors involved in rare spurious correlations and propose mitigation methods accordingly. Specifically, we observe that $\ell_2$ regularization and adding Gaussian noise to inputs can reduce the undesirable effects.

## 1 INTRODUCTION

Neural networks are known to use spurious patterns for classification. Image classifiers use background as a feature to classify objects (Gururangan et al., 2018; Sagawa et al., 2020; Srivastava et al., 2020; Zhou et al., 2021) often to the detriment of generalization (Nagarajan et al., 2020). For example, Sagawa et al. (2020) show that models trained on the Waterbirds datasetcorrelate waterbirds with backgrounds containing water, and models trained on the CelebA dataset Liu et al. (2018) correlate males with dark hair. In all these cases, spurious patterns are present in a substantial number of training points. The vast majority of waterbirds, for example, are photographed next to the water.

Understanding how and when spurious correlations appear in neural networks is a frontier research problem and remains elusive. In this paper, we study spurious correlations in the context where the appearance of spurious patterns is *rare* in the training data. Our motivations are three-fold. First, while it is reasonable to expect that widespread spurious correlations in the training data will be learnt, a related question is what happens when these correlations are *rare*. Understanding if and when they are learnt and how to mitigate them is a first and necessary step before we can understand and mitigate spurious correlations more broadly. Second, rare spurious correlation may inspire us to discover new approaches to mitigate them as traditional approaches such as balancing out groups (Sagawa et al., 2020), subsampling (Idrissi et al., 2021), or data augmentation (Chang et al., 2021) do not apply. Third, rare spurious correlations naturally connect to data privacy. For example, in Leino & Fredrikson (2020), the training set had an image of Tony Blair with a pink background. This led to a classifier that assigned a higher likelihood of the label "Tony Blair" to all images with pink backgrounds. Thus, an adversary could exploit this to infer the existence of "Tony Blair" with a pink background in the training set by by presenting images of other labels with a pink background.

We systematically investigate rare spurious correlations through the following three research questions. First, when do spurious correlations appear, i.e., how many training points with the spurious pattern would cause noticeable spurious correlations? Next, how do rare spurious correlations affect neural networks? Finally, is there any way to mitigate the undesirable effects of rare spurious correlations?

### 1.1 OVERVIEW

We attempt to answer the above questions via both experimental and theoretical approaches. On the experimental side, we introduce spurious correlations into real image datasets by turning a few training data into *spurious examples*, i.e., adding a spurious pattern to a training image from a

target class. We then train a neural network on the modified dataset and measure the strength of the correlation between the spurious pattern and the target class in the network. On the theoretical side, we design a toy mathematical model that enables quantitative analysis on different factors (e.g., the fraction of spurious examples, the signal-to-noise ratio, etc.) of rare spurious correlations. Our responses to the three research questions are summarized in the following.

**Rare spurious correlations appear even when the number of spurious samples is small.** Empirically, we define a *spurious score* to measure the amount of spurious correlations. We find that the spurious score of a neural network trained with only 1 spurious examples out of 60,000 training samples can be significantly higher than that of the baseline. A visualization of the trained model also reveals that the network's weights may be significantly affected by the spurious pattern. In our theoretical model, we further discover that there is a sharp phase transition of spurious correlations from no spurious training example to a non-zero fraction of spurious training examples. Together, these findings provide a strong evidence that spurious correlations can be learnt even when the number of spurious samples is extremely small.

**Rare spurious correlations affect both the privacy and test accuracy.** We analyze the privacy issue of rare spurious correlations via the membership inference attack (Shokri et al., 2017; Yeom et al., 2017), which measures the privacy level according to the hardness of distinguishing training samples from testing samples. We observe that the spurious training examples are more vulnerable to membership inference attacks. That is, it is easy for an adversary to tell whether a spurious sample is from the training set. This apparently raises serious concerns for privacy Leino & Fredrikson (2020) and fairness to small groups Izzo et al. (2021).

We examine the effect of rare spurious correlations on test accuracy through two accuracy notions: the clean test accuracy, which uses the original test examples, and the spurious test accuracy, which adds the spurious pattern to all the test examples. Both empirically and theoretically, we find that clean test accuracy does not change too much while the spurious test accuracy significantly drops in the face of rare spurious correlations. This suggests that the undesirable effect of spurious correlations could be more serious when there is a distribution shift toward having more spurious samples.

**Methods to mitigate the undesirable effects of rare spurious correlations.** Finally, inspired by our theoretical analysis, we examine three regularization methods to reduce the privacy and test accuracy concerns: adding Gaussian noises to the input samples, $\ell_2$ regularization (or equivalently, weight decay), and gradient clipping. We find that adding Gaussian noise and $\ell_2$ regularization effectively reduce spurious score and improve spurious test accuracy. Meanwhile, not all regularization methods could reduce the effects of rare spurious correlations, e.g., gradient clipping. Our findings suggest that rare spurious correlations should be dealt differently from traditional privacy issues. We post it as a future research problem to deepen the understanding of how to mitigate rare spurious correlations.

**Concluding remarks.** The study of spurious correlations is crucial for a better understanding of neural networks. In this work, we take a step forward by looking into a special (but necessary) case of spurious correlations where the appearance of spurious examples is rare. We demonstrate both experimentally and theoretically when and how rare spurious correlations appear and what undesirable consequences are. While we propose a few methods to mitigate rare spurious correlations, we emphasize that there is still a lot to explore, and we believe the study of rare spurious correlations could serve as a guide for understanding the more general cases.

## 2 PRELIMINARIES

We focus on studying spurious correlations in the image classification context. Here, we briefly introduce the notations and terminologies used in the rest of the paper. Let $\mathcal{X}$ be an input space and let $\mathcal{Y}$ be a label space. At the training time, we are given a set of examples $\{(\mathbf{x}_i, y_i)\}_{i \in \{1,\dots,n\}}$ sampled from a distribution $\mathcal{D}_{\text{train}}$, where each $\mathbf{x}_i \in \mathcal{X}$ is associated with a label $y_i \in \mathcal{Y}$. At the testing time, we evaluate the network on test examples drawn from a test distribution. We consider two types of test distribution: the clean test distribution $\mathcal{D}_{\text{ctest}}$ and the spurious test distribution $\mathcal{D}_{\text{stest}}$. Their formal definitions will be mentioned in the related sections.

**Spurious correlation.** A spurious correlation refers to the relationship between two variables in which they are correlated but not causally related. We build on top of the framework used in Nagarajan et al. (2020) to study spurious correlations. Concretely, the input $\mathbf{x}$ is modeled as the output of a

feature map $\Phi_{\mathcal{X}}$ from the feature space $\mathcal{X}_{\text{inv}} \times \mathcal{X}_{\text{sp}}$ to the input space $\mathcal{X}$. Here, $\mathcal{X}_{\text{inv}}$ is the invariant feature space containing the features that causally determine the label and $\mathcal{X}_{\text{sp}}$ which is the spurious feature space that accommodates spurious features. Finally, $\Phi_{\mathcal{X}} : \mathcal{X}_{\text{inv}} \times \mathcal{X}_{\text{sp}} \to \mathcal{X}$ is the function that maps an feature pair $(\mathbf{x}_{\text{inv}}, \mathbf{x}_{\text{sp}})$ to an input $\mathbf{x}$ and $\Phi_{\mathcal{Y}} : \mathcal{X}_{\text{inv}} \to \mathcal{Y}$ is a function that maps the invariant feature $\mathbf{x}_{\text{inv}}$ to a label. Namely, an example $(\mathbf{x}, y)$ is generated by $(\mathbf{x}_{\text{inv}}, \mathbf{x}_{\text{sp}})$ via $\mathbf{x} = \Phi_{\mathcal{X}}(\mathbf{x}_{\text{inv}}, \mathbf{x}_{\text{sp}})$ and $y = \Phi_{\mathcal{Y}}(\mathbf{x}_{\text{inv}})$. Without loss of generality, the zero vector in $\mathcal{X}_{\text{sp}}$, i.e., $0 \in \mathcal{X}_{\text{sp}}$, refers to "no spurious feature" and for any nonzero $\mathbf{x}_{\text{sp}}$ we call $\Phi(\mathbf{x}_{\text{inv}}, \mathbf{x}_{\text{sp}}) - \Phi(\mathbf{x}_{\text{inv}}, 0)$ a spurious pattern. We focus on the case of having a fixed spurious feature $\mathbf{x}_{\text{sp}}$ and leave it as a future direction to study the more general scenarios where there are multiple spurious features.

**Rare spurious correlation.** Following the setting introduced in the previous paragraph, an input distribution $\mathcal{D}$ over $\mathcal{X}$ is induced by a distribution $\mathcal{D}_{\text{feature}}$ over the feature space $\mathcal{X}_{\text{inv}} \times \mathcal{X}_{\text{sp}}$, i.e., to get a sample from $\mathcal{D}$, one first samples $(\mathbf{x}_{\text{inv}}, \mathbf{x}_{\text{sp}})$ from $\mathcal{D}_{\text{feature}}$ and outputs $(\mathbf{x}, y)$ with $\mathbf{x} = \Phi_{\mathcal{X}}(\mathbf{x}_{\text{inv}}, \mathbf{x}_{\text{sp}})$ and $y = \Phi_{\mathcal{Y}}(\mathbf{x}_{\text{inv}})$. Now, we are able to formally discuss the rareness of spurious correlations by defining the spurious frequency of $\mathcal{D}$ as $\gamma := \Pr_{(\mathbf{x}_{\text{inv}}, \mathbf{x}_{\text{sp}}) \sim \mathcal{D}_{\text{feature}}}[\mathbf{x}_{\text{sp}} \neq 0]$.

**Two simple models for spurious correlations.** In general, $\Phi_{\mathcal{X}}$ could be complicated and makes it difficult to detect the appearance of spurious correlations. Here, we consider two simple instantiations of $\Phi_{\mathcal{X}}$ and demonstrate that undesirable learning outcomes already appear even in these simplified settings. First, the *overlapping model* (used in Sec. 3) where the spurious feature is put on top of the invariant feature, i.e., $\mathcal{X} = \mathcal{X}_{\text{inv}} = \mathcal{X}_{\text{sp}}$ and $\Phi_{\mathcal{X}}(\mathbf{x}_{\text{inv}}, \mathbf{x}_{\text{sp}}) = \mathbf{x}_{\text{inv}} + \mathbf{x}_{\text{sp}}$ or $\Phi_{\mathcal{X}}(\mathbf{x}_{\text{inv}}, \mathbf{x}_{\text{sp}}) = clip(\mathbf{x}_{\text{inv}} + \mathbf{x}_{\text{sp}})$ where $clip$ is a function that truncates an input pixel when its value exceeds a certain range. Second, the *concatenate model* (used in Sec. 5) where the spurious feature is concatenated to the invariant feature, i.e., $\mathcal{X} = \mathcal{X}_{\text{inv}} \times \mathcal{X}_{\text{sp}}$ and $\Phi_{\mathcal{X}}(\mathbf{x}_{\text{inv}}, \mathbf{x}_{\text{sp}}) = (\mathbf{x}_{\text{inv}}, \mathbf{x}_{\text{sp}})$.

## 3 RARE SPURIOUS CORRELATIONS ARE LEARNT BY NEURAL NETWORKS

We start with an empirical study of rare spurious correlations in neural networks. We train a neural network using a modified training dataset given by the *overlapping model* where a spurious pattern is added to a few training examples with the same label (target class). We then analyze the effect of these spurious training examples through three difference angles: (i) a quantitative analysis on the appearance of spurious correlations via an empirical measure, *spurious score*, (ii) a qualitative analysis on the appearance of spurious correlations through visualizing the network weights, and (iii) an analysis on the consequences of rare spurious correlations in terms of privacy and test accuracy.

### 3.1 INTRODUCING SPURIOUS EXAMPLES TO NETWORKS

As we don't have access to the underlying ground-truth feature of an empirical data, we artificially introduce spurious features into the training dataset. Concretely, given a dataset (e.g., MNIST), we treat each training example $\mathbf{x}$ as an invariant feature. Next, we pick a target class $c_{\text{tar}}$ (e.g., the zero class), a spurious pattern $\mathbf{x}_{\text{sp}}$ (e.g., a yellow square at the top-left corner), and a mapping $\Phi_{\mathcal{X}}$ that combines a training example with the spurious pattern. Finally, we randomly select $n$ training examples $\mathbf{x}_1, \ldots, \mathbf{x}_n$ from the target class $c_{\text{tar}}$ and replace these examples with $\Phi_{\mathcal{X}}(\mathbf{x}_i, \mathbf{x}_{\text{sp}})$ for each $i = 1, \ldots, n$. See Fig. 1 and the following paragraphs for a detailed specification of our experiments.

**Datasets & the target class $c_{\text{tar}}$.** We consider three commonly used image datasets: MNIST (LeCun, 1998), Fashion (Xiao et al., 2017), and CIFAR10 (Krizhevsky & Hinton, 2009). MNIST and Fashion have $60,000$ training examples, and CIFAR10 has $50,000$. We set the first two classes of each dataset as the target class ($c_{\text{tar}} = \{0, 1\}$), which are zero and one for MNIST, T-shirt/top, and trouser for Fashion, and airplane and automobile for CIFAR10. See App. D.1 for more experimental details.

**Spurious patterns $\mathbf{x}_{\text{sp}}$.** We consider seven different spurious patterns (Fig. 1) for this study. The patterns *small 1 (S1)*, *small 2 (S2)*, and *small 3 (S3)* are designed to test if a neural network can learn the correlations between small patterns and the target class. The patterns *random 1 (R1)*, *random 2 (R2)*, and *random 3 (R3)* are patterns with each pixel value being uniformly random sampled from $[0, r]$, where $r = 0.25, 0.5, 1.0$ (we sample the pattern once and fix it throughout each experiment). We study whether a network learns to correlate random noise with a target class. In addition, by comparing random patterns with these small patterns, we can understand the impact of localized and dispersed spurious patterns. Lastly, the pattern *core (Inv)* is designed for MNIST with $c_{\text{tar}} = 0$ to understand what would happen if the spurious pattern overlaps with the core feature of another class.

**The choice of the combination function $\Phi_{\mathcal{X}}$.** The function $\Phi_{\mathcal{X}}$ combines the original example $\mathbf{x}$ with the spurious pattern $\mathbf{x}_{\mathrm{sp}}$ into a spurious example. For simplicity, we consider the *overlapping model* where $\Phi_{\mathcal{X}}$ directly adds the spurious pattern $\mathbf{x}_{\mathrm{sp}}$ onto the original example $\mathbf{x}$ and then clips the value of each pixel to $[0, 1]$, i.e., $\Phi_{\mathcal{X}}(\mathbf{x}, \mathbf{x}_{\mathrm{sp}}) = clip_{[0,1]}(\mathbf{x} + \mathbf{x}_{\mathrm{sp}})$.

**The number of spurious examples.** For MNIST and Fashion, we randomly insert the spurious pattern to $n = 0, 1, 3, 5, 10, 20, 100, 2000$, and $5000$ training examples labeled as the target class $c_{\mathrm{tar}}$. These training examples inserted with a spurious pattern are called spurious examples. For CIFAR10, we consider datasets with $n = 0, 1, 3, 5, 10, 20, 100, 500$, and $1000$ spurious examples. Note that $0$ spurious example means the original training set is not modified.

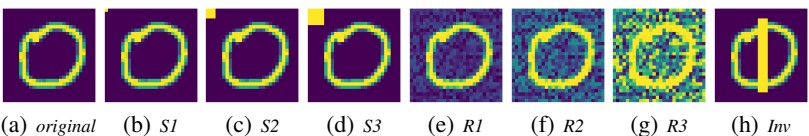

(a) *original*    (b) *S1*    (c) *S2*    (d) *S3*    (e) *R1*    (f) *R2*    (g) *R3*    (h) *Inv*

Figure 1: Different spurious patterns considered in the experiment.

## 3.2 QUANTITATIVE ANALYSIS: SPURIOUS SCORE

To evaluate the strength of spurious correlations in a neural network, we design an empirical quantitative measure, *spurious score*, as follows. Let $f_c(\mathbf{x})$ be the neural network's predicted probability of an example $\mathbf{x}$ belonging to class $c$. Intuitively, the larger the *prediction difference* $f_{c_{\mathrm{tar}}}(\mathbf{x}) - f_{c_{\mathrm{tar}}}(\Phi_{\mathcal{X}}(\mathbf{x}, \mathbf{x}_{\mathrm{sp}}))$ is, the stronger spurious correlations the neural network $f$ had learned. To quantify the effect of spurious correlations, we measure how frequently the prediction difference of the test examples exceed a certain threshold. Formally, let $\epsilon > 0$, we define the $\epsilon$-*spurious score* as the fraction of test example $\mathbf{x}$ that satisfies

$$f_{c_{\mathrm{tar}}}(\Phi_{\mathcal{X}}(\mathbf{x}, \mathbf{x}_{\mathrm{sp}})) - f_{c_{\mathrm{tar}}}(\mathbf{x}) > \epsilon. \tag{1}$$

In other words, spurious score measures the portion of test examples that get an non-trivial increase in the predicted probability of the target class $c_{\mathrm{tar}}$ when the spurious pattern is presented.

We make three remarks on the definition of spurious score. First, as we don't have any prior knowledge on the structure of $f$, we use the fraction of test examples satisfying Eq. (1) as opposed to other function of $f_{c_{\mathrm{tar}}}(\Phi_{\mathcal{X}}(\mathbf{x}, \mathbf{x}_{\mathrm{sp}})) - f_{c_{\mathrm{tar}}}(\mathbf{x})$ (e.g., taking average) to avoid non-monotone or unexplainable scaling. Second, the choice of the threshold $\epsilon$ is to avoid numerical errors to affect the result. In our experiment, we pick $\epsilon = 1/(\#\text{classes})$ (e.g., in MNIST we pick $\epsilon = 1/10$) and empirically similar conclusions can be made with other choices of $\epsilon$. Finally, we point out that spurious score captures the privacy concern raised by the "Tony Blair" example mentioned in the introduction.

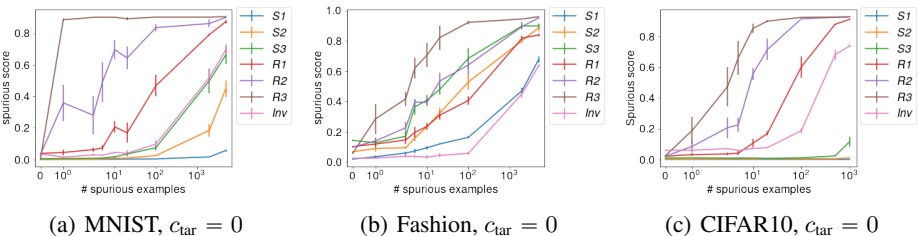

(a) MNIST, $c_{\mathrm{tar}} = 0$    (b) Fashion, $c_{\mathrm{tar}} = 0$    (c) CIFAR10, $c_{\mathrm{tar}} = 0$

Figure 2: Each figure shows the mean and standard error of the spurious scores on three datasets, MNIST, Fashion, and CIFAR10, $c_{\mathrm{tar}} = 0$, and different numbers of spurious examples.

**Empirical findings.** We repeat the measurement of spurious scores on five neural networks trained with different random seeds. Fig. 2 shows the spurious scores for each dataset and pattern as a function of the number of spurious examples. Starting with the random pattern *R3*, we see that the spurious scores increase significantly from zero to three spurious examples in all six cases (three datasets and two target classes). This shows that neural networks can learn rare spurious correlations with as little as *one to three spurious examples*. Since all three datasets have $50,000$ or more training examples, it is surprising that the networks learn a strong correlation with extremely small amount of spurious examples. The result for other $c_{\mathrm{tar}}$ are similar and can be found in App. D.

A closer look at Fig. 2 reveals a few other interesting observations. First, comparing the small and random patterns, we see that random patterns generally have a higher spurious score. This suggests that dispersed patterns that are spread out over multiple pixels may be more easily learnt than more concentrated ones. Second, spurious correlations are learnt even for *Inv*, on $c_{\text{tar}} = 0$ and MNIST (recall that *Inv* is designed to be similar to the core feature of class one.) This suggests that spurious correlations may be learnt even when the pattern overlaps with the foreground. Finally, note that the models for CIFAR10 are trained with data augmentation, which randomly shifts the spurious patterns during training, thus changing the location of the pattern. This suggests that these patterns can be learnt regardless of data augmentation.

In App. E.1, we also conduct a qualitative study by visualizing the weights of networks trained with rare spurious examples. We find that the spurious pattern can leave a trace on the weights of the neural network. This observation strengthens our claim that neural networks are significantly effected by rare spurious examples.

## 3.3 NATURAL RARE SPURIOUS CORRELATIONS

A question is whether rare spurious correlations are also learnt on real (natural) data. To answer this question, we conduct an experiment focusing on natural spurious patterns using the NICO++ (Zhang et al., 2022) dataset, which is designed for studying non-I.I.D. image classification There are two labels of each image in the dataset: an object class (e.g., airplane) and the context (e.g., autumn). The context can then serve as a source of spurious features: if a context only appears with the same class during the training stage (e.g., autumn context only shows up when the concept is airplane), then the algorithm might think the context is causally related to the classification of the object class (e.g., classifying a bear in the autumn context as an airplane).

The NICO++ dataset consists of 55838 training images, sixty classes and six contexts, including autumn, dim, grass, outdoor, rock, and water. We split the dataset seven to three as the training and testing set. For each experiment, we use an object-context pair as an invariant-spurious pair for studying rare spurious correlations. For one trial of our experiment, we pick a context (i.e., a spurious feature) and remove all the appearance of this context in the training examples. To introduce spurious training examples, we select an object class as the target class and add a number of examples that are labeled with the target class and this context. We use the ImageNet pretrained ResNet50 from `torchvision` and train twenty epochs on this modified training set. During testing, we collect all testing examples that do not belong to the spurious class but are under the spurious context and measure how many of these examples are predicted as the spurious class.

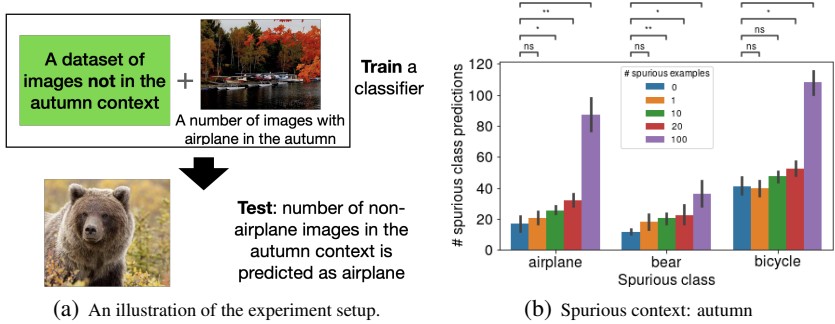

(a) An illustration of the experiment setup.   (b) Spurious context: autumn

Figure 3: (b) The number of non-spurious test examples that get predicted as the spurious class. We conduct Welch's t-test (Welch, 1947) on the number of spurious class predictions between the model trained without spurious examples and models trained with different number of spurious examples. The notations for the p-values: ns: $0.05 < p \leqslant 1.$, *: $10^{-2} < p \leqslant 0.05$, **: $10^{-3} < p \leqslant 10^{-2}$, ***: $10^{-4} < p \leqslant 10^{-3}$, ****: $p \leqslant 10^{-4}$.

**Results.** The experiments are repeated five times with different numbers of spurious examples in the training set and the results are in Fig. 3(b). In App. E.5, we show the results of two other spurious contexts, dim and grass, which also supports our conclusion. in total, we run a total of nine trials and five of them are significantly affected by just ten spurious training examples (which is less than 0.02% in the whole training examples). This suggests that rare spurious correlations also occur when the spurious features are natural.

# 4 CONSEQUENCES OF RARE SPURIOUS CORRELATIONS

In the previous analysis, we demonstrated that spurious correlations appear quantitatively and qualitatively in neural networks even when the number of spurious examples is small. Now, we investigate the potentially undesirable effects through the lens of privacy and test accuracy. In this section, the results for MNIST are similar to Fashion and CIFAR10 and are deferred to App. D.

**Privacy.** We evaluate the privacy of a neural network (the target model) through membership inference attack. We follow the setup for black-box membership inference attack (Shokri et al., 2017; Yeom et al., 2017). We record how well an attack model can distinguish whether an example is from the training or testing set using the output of the target model (equivalently to a binary classification problem). If the attack model has a high accuracy, this means that the target model is leaking out information from the training (private) data. The experiment is repeated ten times with their test accuracy recorded. For more detailed setup, please refer to App. D.2.

*Results on membership inference attack.* Fig. 4 shows the mean and standard error of the attack model's test accuracy on all test examples and spurious examples. We see that the accuracies on spurious examples is generally higher when the number of spurious examples are small, which means that spurious examples are more vulnerable to membership inference attacks when appeared rarely. Although membership inference attack is a different measure for privacy than spurious score, it can be a corroboration evidence that supports the fact that privacy is leaked from spurious examples.

**Test accuracy.** We measure two types of test accuracy on neural networks trained on different number of spurious examples. The *clean test accuracy* measures the accuracy of the trained model on the original test data. The *spurious test accuracy* simulates the case where there is a distribution shift during the test time. Formally, spurious test accuracy is defined as the accuracy on a new test dataset constructed by adding spurious features to all the test examples with a label different from $c_{\text{tar}}$.

*Results on clean test accuracy.* We observe that the change in clean test accuracy in our experiments is small. Across all the models trained in Fig. 2, the minimum, maximum, average, and standard deviation of the test accuracy for each dataset are: MNIST: (.976, .983, .980, .001), Fashion: (.859, .903, .890, .010), and CIFAR10: (.876, .893, .886, .003).

*Results on spurious test accuracy.* The results are shown in Fig. 5. We have two observations. First, we see that there are already some accuracy drop even when spurious test accuracy is evaluated on models trained on zero spurious examples. This means that these models are not robust to the existence of spurious features. This phenomena is prominent for spurious patterns with larger norm such as *R3*. Second, we see that spurious test accuracies start to drop even more at around 10 to 100 spurious examples. This indicates that even with .01 % to .001 % of the overall training data filled with spurious examples of a certain class, the robustness to spurious features can drop significantly.

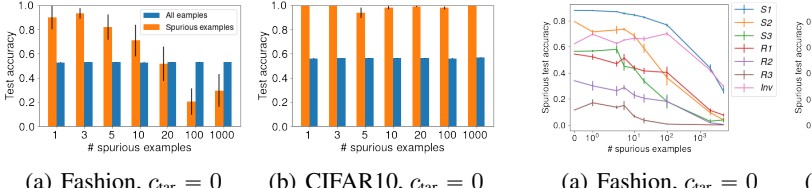

| (a) Fashion, $c_{\text{tar}} = 0$ | (b) CIFAR10, $c_{\text{tar}} = 0$ | (a) Fashion, $c_{\text{tar}} = 0$ | (b) CIFAR10, $c_{\text{tar}} = 0$ |

Figure 4: The test accuracy of the membership inference attack model on all examples vs. spurious examples. See App. E.3 for all results.

Figure 5: The mean and standard error of the spurious test accuracy under different number of spurious examples. See App. E.4 for all results.

**Discussion.** Our experimental results suggest that neural networks are *highly* sensitive to very small amounts of spurious training data. Furthermore, the learnt rare spurious correlations cause undesirable effects on privacy and test accuracy. Easy learning of rare spurious correlations can lead to privacy issues (Leino & Fredrikson, 2020) – where an adversary may infer the presence of a confidential image in a training dataset based on output probabilities. It also raises fairness concerns as a neural network can draw spurious conclusions about a minority group if a small number of subjects from this group are present in the training set (Izzo et al., 2021). We recommend to test and audit neural networks thoroughly before deployment in these applications.

Beyond the empirical analysis explained in this section, we also explore how other factors such as the strength of spurious patterns, network architectures, and optimization methods, affect spurious correlations. We find that one cannot remove rare spurious correlations by simply tuning these parameters. We also observe that neural networks with more parameters may not always learn spurious correlations more easily, which is counter to Sagawa et al. (2020)'s observation. The detailed results and discussions are provided in App. B.

## 5 THEORETICAL UNDERSTANDING

In this section, we devise a mathematical model to study rare spurious correlations. The theoretical analysis not only provides an unifying understanding to explain the experimental findings but also inspires us to propose methods to reduce the undesirable effects of rare spurious correlations in Sec. 6. We emphasize that the purpose of the theoretical analysis is to capture the key factors in rare spurious correlations and we leave it as a future research direction to further deepen the theoretical study.

To avoid unnecessary mathematical complications, we make two simplifications in our theoretical analysis: (i) we focus on the *concatenate model* and (ii) the learning algorithm is linear regression with mean square loss. For (i), we argue that this is the simplest scenario of spurious correlations and hence it is a necessary step before we understand general spurious correlations. While the experiments in Sec. 3 work in the *overlapping model*, we believe that the high level messages of our theoretical analysis would extend to there as well as other more general scenarios. For (ii), we pick a simpler learning algorithm in order to have an analytical characterization of the algorithm's performance. This is because we aim to have an understanding of how the different factors (e.g., the fraction of spurious inputs, the strength of spurious feature, etc.) of spurious correlations play a role.

### 5.1 A THEORETICAL MODEL TO STUDY RARE SPURIOUS CORRELATIONS

We consider a binary classification task to model the appearance of rare spurious correlations. Let $\mathcal{X} = \mathcal{X}_{\text{inv}} \times \mathcal{X}_{\text{sp}}$ be an input vector space and let $\mathcal{Y} = \{-1, 1\}$ be a label space. Let $\gamma \in [0, 1]$ be the parameter for the fraction of spurious samples, let $\mathbf{x}_-, \mathbf{x}_+ \in \mathcal{X}_{\text{inv}}$ be the invariant features of the two classes, let $\mathbf{x}_{\text{sp}} \in \mathcal{X}_{\text{sp}}$ be the spurious feature, and let $\sigma_{\text{inv}}^2, \sigma_{\text{sp}}^2 > 0$ be the parameters for the variance along $\mathcal{X}_{\text{inv}}$ and $\mathcal{X}_{\text{sp}}$ respectively. Finally, the target class is +, i.e., $c_{tar} = +$. We postpone the formal definitions of the training distribution $\mathcal{D}_{\text{train}}$, the clean text distribution $\mathcal{D}_{\text{ctest}}$, and the spurious test distribution $\mathcal{D}_{\text{stest}}$ to App. A. See also Fig. 6 for some pictorial examples.

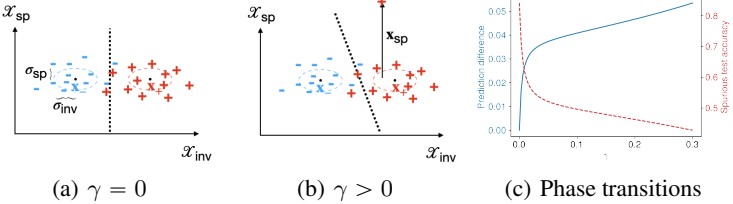

| (a) $\gamma = 0$ | (b) $\gamma > 0$ | (c) Phase transitions |

Figure 6: Examples of the training distribution $\mathcal{D}_{\text{train}}$ and phase transitions in our theoretical model. (a)-(b) With equal probability a training example is sampled from either $\mathcal{N}(\mathbf{x}_+, \sigma_{\text{inv}} I_{\text{inv}} + \sigma_{\text{sp}} I_{\text{sp}})$ or $\mathcal{N}(\mathbf{x}_-, \sigma_{\text{inv}} I_{\text{inv}} + \sigma_{\text{sp}} I_{\text{sp}})$. With probability $\gamma$, a + sample will be concatenated with the spurious pattern $\mathbf{x}_{\text{sp}}$. The dotted line is the decision boundary of the optimal classifier. (c) Both the spurious test accuracy and the prediction difference exhibit a phase transition at $\gamma = 0$.

### 5.2 ANALYSIS FOR LINEAR REGRESSION WITH MEAN SQUARE LOSS AND $\ell_2$ REGULARIZATION

In this paper, we analyze our theoretical model in the setting of linear regression with $\ell_2$ loss. We analytically derive the test accuracy and the prediction difference $f_{c_{\text{tar}}}(\Phi_{\mathcal{X}}(\mathbf{x}, \mathbf{x}_{\text{sp}})) - f_{c_{\text{tar}}}(\mathbf{x})$ in App. A and here we present our observations. Here, we study the prediction difference as a proxy for the spurious score since the latter is always either 0 or 1 for a linear classifier under our model.

**Observation 1: A phase transition of spurious test accuracy and prediction difference at $\gamma = 0$.** Our theoretical analysis first suggests that there is a phase transition of the spurious test accuracy and the prediction difference spurious score at $\gamma = 0$, i.e., there is an sharp growth/decay within an interval near $\gamma = 0$. The phase transition matches our previous experimental studies discussed

in Sec. 3. This indicates that the effect of spurious correlations is spontaneous rather than gradual. To be more quantitative, the phase transition takes place when the signal-to-noise ratio of spurious feature, i.e., $\|\mathbf{x}_{sp}\|_2^2/\sigma_{sp}^2$, is large (see App. A for details). This further suggests us to increase the variance in the spurious dimension and leads to our next two observations.

**Observation 2: adding Gaussian noises lowers spurious score.** The previous observation on the importance of spurious signal-to-noise ratio $\|\mathbf{x}_{sp}\|_2^2/\sigma_{sp}^2$ immediately suggests us to add Gaussian noises to the input data to *lower* $\|\mathbf{x}_{sp}\|_2^2/\sigma_{sp}^2$. Indeed, the prediction difference becomes smaller in most parameter regimes, however, both the clean test accuracy and the spurious test accuracy decrease. Intuitively, the effect of adding noises is to mix the invariant feature with the spurious feature and the decrease of test accuracy is as expected. Thus, to simultaneously lower prediction difference and improve test accuracy, one needs to detect the spurious feature in some ways.

**Observation 3: $\ell_2$ regularization improves test accuracy and lowers spurious score.** Finally, our theoretical analysis reveals that there are two parameter regimes where adding $\ell_2$ regularization to linear regression can improves accuracy and lowers the prediction difference. First, when $\sigma_{inv}^2$ is small, $\gamma$ is small, and the spurious signal-to-noise ratio $\|\mathbf{x}_{sp}\|_2^2/\sigma_{sp}^2$ is large. Second, when $\sigma_{inv}^2$ is large and both $\gamma$ and $\|\mathbf{x}_{sp}\|_2^2/\sigma_{sp}^2$ are mild. Intuitively, $\ell_2$ regularization suppresses the use of features that only appears on a small number of training examples.

**Discussion.** Our theoretical analysis quantitatively demonstrates the phase transition of spurious correlations at $\gamma = 0$ and the importance of the spurious signal-to-noise ratio $\|\mathbf{x}_{sp}\|_2^2/\sigma_{sp}^2$. This not only coincides with our empirical observation in Sec. 3 but also suggests future directions to mitigate rare spurious correlations. Specifically, one general approach to reduce the undesirable effects of rare spurious correlations would be designing learning algorithms that projects the input into a feature space that has a low spurious signal-to-noise ratio.

## 6 MITIGATION OF RARE SPURIOUS CORRELATION

Prior work uses group rebalancing (Idrissi et al., 2021; Sagawa et al., 2020; 2019; Kulynych et al., 2022), data augmentation (Chang et al., 2021) or learning invariant classifier (Arjovsky et al., 2019) to mitigate spurious correlations. However, these methods usually requires additional information on what the spurious feature is, and in rare spurious correlations, identifying the spurious feature can be hard. Thus, we may require different techniques.

Our theoretical result suggest that $\ell_2$ regularization (weight decay) and adding Gaussian noise to the input (noisy input) may reduce the degree of spurious correlation being learnt. In addition, we examine an extra regularization method – gradient clipping.

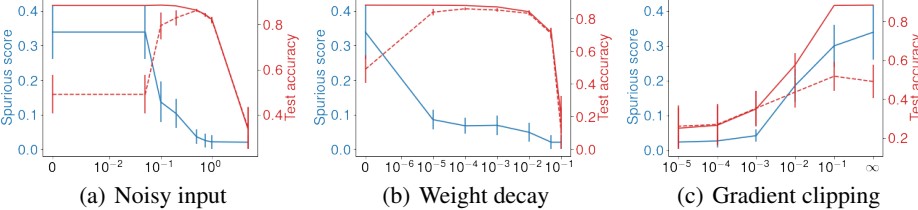

| (a) Noisy input | (b) Weight decay | (c) Gradient clipping |

Figure 7: Spurious score (solid blue line), clean test accuracy (solid red line), and spurious test accuracy (dotted red line) vs. the regularization strength on Fashion with different regularization methods. For the experiment, we fix the spurious pattern to be *S3* and the target class $c_{tar} = 0$. We compute the average spurious score and clean test accuracy across models trained with 1, 3, 5, 10, 20, and 100 spurious examples and five random seeds. The results for MNIST and CIFAR10 and spurious pattern *R3* are in App. D, which shows similar results.

**Results.** The results are shown in Fig. 7. We see that with a properly set regularization strength for noisy input and weight decay, one may reduce the spurious score and increase spurious test accuracy without sacrificing much accuracy. This significantly reduces the undesirable consequences brought by rare spurious correlations. This aligns with our observation in the theoretical model and suggests that neural networks may share a similar property with linear models. We also find that gradient clipping cannot mitigate spurious correlation without reducing the test accuracy. Finally, we observe that all these methods are unable to completely avoid learning spurious correlations.

**Data deletion methods.** Another idea for mitigating rare spurious correlation is to apply data deletion methods (Izzo et al., 2021). In App. C, we experiment with two data deletion methods, incremental retraining and group influence functions (Basu et al., 2020a). However, they are not effective.

**Discussion.** Regarding mitigating rare spurious correlations, a provable way to prevent learning them is differential privacy (Dwork et al., 2006), which ensures that the participation of a single person (or a small group) in the dataset does not change the probability of any classifier by much. This requires noise addition during training, which may lead to a significant loss in accuracy (Chaudhuri et al., 2011; Abadi et al., 2016). If we know which are the spurious examples, then we can remove spurious correlations via an indistinguishable approximate data deletion method (Ginart et al., 2019; Neel et al., 2020); however, these methods provide lower accuracy for convex optimization and have no performance guarantees for non-convex. An open problem is to design algorithms or architectures that can mitigate these without sacrificing prediction accuracy.

Prior work (Sagawa et al., 2019; Kulynych et al., 2022) suggests that proper use of regularization methods plus well-designed loss functions can mitigate some types of spurious correlations. However, these regularization methods are used either in an ad-hoc manner or may reduce test accuracy. As shown in our experiment, not all regularization methods can remove rare spurious correlations without reducing the accuracy. This suggests that different regularization methods may be specifically tied to be able to mitigate certain kinds of spurious correlation. The exact role that regularization methods play in reducing spurious correlations is still an open question. Figuring out what kinds of spurious correlations can be mitigated by which regularization methods is an interesting future direction.

## 7 RELATED WORK

**Spurious correlations.** Previous work has looked at spurious correlations in neural networks under various scenarios, including test time distribution shift (Sagawa et al., 2020; Srivastava et al., 2020; Bahng et al., 2020; Zhou et al., 2021; Khani & Liang, 2021), confounding factors in data collection (Gururangan et al., 2018), the effect of image backgrounds (Xiao et al., 2020), and causality (Arjovsky et al., 2019). However, in most works, spurious examples often constitute a significant portion of the training set. In contrast, we look at spurious correlations introduced by a small number of examples (rare spurious correlations). Concurrent work (Hartley & Tsaftaris, 2022) measures spurious correlation caused by few examples. However, they did not show the consequences of these spurious correlations nor discuss ways to mitigate them.

**Memorization in neural networks.** Prior work has investigated how neural networks can inadvertently memorize training data (Arpit et al., 2017; Carlini et al., 2019; 2020; Feldman & Zhang, 2020; Leino & Fredrikson, 2020). Methods have also been proposed to measure this kind of memorization, including the use of the influence function (Feldman & Zhang, 2020) and likelihood estimates (Carlini et al., 2019). Our work focuses on partial memorization instead of memorizing individual examples, and our proposed method may be potentially applicable in more scenarios.

A line of work in the security literature exploits the memorization of certain patterns to compromise neural networks. The backdoor attack from Chen et al. (2017) attempts to change hard label predictions and accuracy by inserting carefully crafted spurious patterns. Sablayrolles et al. (2020) design specific markers that allow adversaries to detect whether images with those particular markers are used for training in a model. Another line of research on data poisoning attack Xiao et al. (2015); Wang & Chaudhuri (2018); Burkard & Lagesse (2017) aims to degrade the performance of a model by carefully altering the training data. In contrast, our work looks at rare spurious correlations from *natural spurious patterns*, instead of adversarially crafted ones. App. F has detailed discussions.

## 8 CONCLUSION

The learning of spurious correlation is a complex process, and it can have unintended consequences. As neural networks are getting more widely applied, it is crucial to better understand spurious correlations. There are many open questions remain. For example, besides the distribution shift that adds spurious features, are there any other types of distribution shift that will affect the accuracy? Another limitation of our current study is that our experiments are conducted only on image classification tasks, which may not generalize to others.

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
