# OpenReview forum: "Understanding Rare Spurious Correlations in Neural Networks"
_ICLR.cc/2023/Conference — Submitted to ICLR 2023_

### Official Review · Reviewer_yphb · 2022-10-14

**Confidence:** 3
**Clarity, Quality, Novelty And Reproducibility:** Please see **Strength And Weaknesses**.
**Correctness:** 2
**Technical Novelty And Significance:** 2
**Empirical Novelty And Significance:** 2
**Recommendation:** 5

**Strength And Weaknesses:**

**Strength**

1.Studying the relation between the number of spurious examples and the strength of spurious correlations is an important problem. The experiments show that neural nets could learn natural rare spurious correlations are new and interesting.

2.This paper is well-written. Extensive results are well organized.

3.The analysis on the privacy risk of spurious examples is new.

**Weaknesses**

1.The spurious features in Section 3.1 and 3.2 are very similar to backdoor triggers. They both are some artificial patterns that only appear a few times in the training set. For example, Chen et al. (2017) use random noise patterns. Gu et al. (2019) [1] use single-pixel and simple patterns as triggers. It is well-known that a few training examples with such triggers (rare spurious examples in this paper) would have a large impact on the trained model.

2.How neural nets learn natural rare spurious correlations is unknown to the community (to the best of my knowledge). However, most of analysis and ablation studies use the artificial patterns instead of natural spurious correlations. Duplicating the same artificial pattern for multiple times is different from natural spurious features, which are complex and different in every example.

3.What’s the experiment setup in Section 3.3? (data augmentation methods, learning rate, etc.).

[1]: BadNets: Evaluating Backdooring Attacks on Deep Neural Networks. https://messlab.moyix.net/papers/badnets_ieeeaccess19.pdf

**Summary Of The Paper:**

This paper studies how different numbers of spurious training examples (from one to thousands) affect neural nets. They measure how the predictions of test examples change when the spurious features are added. They show even a few spurious training examples can make the model outputs be affected by the spurious feature.

The authors consider two different types of spurious features. The first one is some artificial patterns, e.g., a square of yellow pixels. The second one is some natural spurious features in real data. They use the NICO++ dataset that has different backgrounds (grass, rock, water..) for each class. The authors then study the consequences of learning rare spurious features on privacy and accuracy. They also analyze the findings with a toy theoretical model and study several simple mitigations.


**Summary Of The Review:**

Given that 1) the findings about artificial spurious features seem incremental because of the backdoor attack literature; 2) how neural nets learn natural rare spurious features is not well studied. I'm leaning to reject this paper.

---

> ### Author Response · Authors · 2022-11-08
> **Response**
>
> - The spurious features in Section 3.1 and 3.2 are very similar to backdoor triggers. They both are some artificial patterns that only appear a few times in the training set. For example, Chen et al. (2017) use random noise patterns. Gu et al. (2019) [1] use single-pixel and simple patterns as triggers. It is well-known that a few training examples with such triggers (rare spurious examples in this paper) would have a large impact on the trained model.
>
> We agree that prior works on backdoor triggers do study the case where artificial patterns only appear a few times in the training set. However, we would like to point out that there is a major difference in the number of spurious examples needed, which leads to very different consequences in terms of privacy. For example, Chen et al. (2017) showed that they would need to inject 50 examples for the inject pattern-key attack to work, and Gu et al. (2019) [1] require 6000 (10%) spurious examples. In contrast, in our work, we focus on the case of 1 to 10 spurious examples. This crucial difference leads us to discover the concrete privacy concern (one example affected by spurious features is much more vulnerable than 50 examples in terms of privacy).
>
> - How neural nets learn natural rare spurious correlations is unknown to the community (to the best of my knowledge). However, most of analysis and ablation studies use the artificial patterns instead of natural spurious correlations. Duplicating the same artificial pattern for multiple times is different from natural spurious features, which are complex and different in every example.
>
> We agree that natural rare spurious correlations have not been well studied in the past, and we point out that one of our major contributions is to demonstrate that natural rare spurious correlations do exist. There are two main reasons why we focus on the analysis of artificial patterns. First, we currently don’t even have a good understanding of rare spurious correlations in simple cases. For example, even in the simple case of artificial patterns, how to completely mitigate rare spurious correlations is still unknown. Directly jumping into the complex case of natural rare spurious correlation would not provide useful insight. Second, natural spurious correlation can be much more complex. One major hurdle is that on the NICO++ dataset, we cannot inject spurious features easily. For example, changing the background of an image from autumn to grass cannot be executed cleanly without introducing other confounding factors. Together with these reasons, we suggest getting a solid understanding of the simple case as we did in our paper first before gradually moving towards a more complex scenario.
>
> In addition, we believe our experiments in Section 3 are enough to support our claim that natural rare spurious correlations do occur. It would be great if you could point out some confounding factors that we may have missed and would require an ablation study on.
>
> - What’s the experiment setup in Section 3.3? (data augmentation methods, learning rate, etc.).
>
> We apologize for missing these details in the paper. We use the ImageNet pretrained ResNet50 from torchvision and train twenty epochs on the modified NICO++ dataset. We set the batch size to 32 and use the SGD optimizer with a learning rate of 0.01 and momentum of 0.9. For data augmentation, during training, we resize each image to 256 pixels on the short side of the image, random square crop the resized image to 224x224 pixels, random horizontally flip the cropped image, and normalize the image by minus the mean ([0.485, 0.456, 0.406]) and divide by the standard deviation ([0.229, 0.224, 0.225]) of the pre-trained dataset. During testing, we resize the image to 256 pixels on the short side of the image, center square crop the resized image to 224x224 pixels, and normalized the image the same way as in the training time. We will add this piece of information to the final version.

---

> > ### Comment · Reviewer_yphb · 2022-11-27
> > **Reply from Reviewer yphb**
> >
> >
> > Thank you for your response.
> >
> > 1. Regards to prior works on backdoor triggers.
> >
> > I agree that there is a difference in the number of spurious examples, e.g., 1\~10 vs 50. However, given that 50 spurious examples can perform a strong backdoor attack, I'm not surprised that 1\~10 spurious examples would have some weaker influence on the trained model.
> >
> > In Appendix F, where you discuss the difference between this work and backdoor attack, you write "First, the spurious examples are not adversarial and, most of the time, are natural and simple." I don't think the spurious patterns in Figure 1 are natural. I would suggest the authors to explain more on this claim.
> >
> > 2. Regards to natural rare spurious correlations.
> >
> > Thank you for the experiment details. I would suggest the authors to run ablation studies with larger learning rates and different data augmentations. This is because prior works show 1) different learning rates bias to different types of features [1]; 2) strong data augmentations reduce the influence of spurious features [2].
> >
> >
> > [1]: Towards Explaining the Regularization Effect of Initial Large Learning Rate in Training Neural Networks, https://arxiv.org/abs/1907.04595.
> >
> > [2]: Unlearnable Examples: Making Personal Data Unexploitable, https://arxiv.org/abs/2101.04898.

---

> > > ### Author Response · Authors · 2022-11-28
> > > **Response to Reviewer yphb**
> > >
> > > Thank you for the response.
> > >
> > > 1. Regards to prior works on backdoor triggers.
> > >
> > > For our claim in Appendix F, we were referring to the examples with natural spurious patterns (in Section 3.3 and Figure 3) while we also looked into non-natural spurious pattern (e.g., Figure 1) for the completeness of the study.  To the best of our knowledge, there has been no prior work showing natural spurious pattern in such a small number of spurious examples.
> > >
> > > 2. Regards to natural rare spurious correlations.
> > >
> > > We agree with the idea of running our experiments on different choices of learning rate as an interesting future work. Indeed, understanding the effects of various training parameters on rare spurious correlations is of great interest and importance. Nonetheless, as we initiated the study of rare spurious correlations in this paper, the main focus is on (i) demonstrating the appearances and effects of rare spurious correlations in various settings, (ii) a theoretical model to explain the qualitative and quantitative scaling of accuracy and privacy, and (iii) several ways to mitigate rare spurious correlations.
> > >
> > > We would also like to emphasize that such privacy threat operates in the case where we don't know what is the spurious feature and which examples are spurious (if we know such information, we should just take out these examples from the training set). Therefore, the methods used in [2] is not available because we do not know which example to make unlearnable.

---

### Official Review · Reviewer_ajJR · 2022-10-22

**Confidence:** 4
**Clarity, Quality, Novelty And Reproducibility:** The paper is very clear and of good q…
**Correctness:** 3
**Technical Novelty And Significance:** 4
**Empirical Novelty And Significance:** 4
**Recommendation:** 6

**Strength And Weaknesses:**

Strength:
- Extensive experiments are performed across datasets, the strength of spurious patterns, network architectures, and optimization methods, showing that even a small amount of spurious correlation will also influence the model performance.
- Theoretical analysis of a simple but meaningful model is conducted, moreover, inspired by the analysis, several techniques are proposed and validated with experiments.

Weaknesses:
- Does the technique proposed also work when the majority of the data present spurious correlation, e.g. Adding l2 regularization? Because as mentioned in the paper: l2 regularization only "suppresses the use of features that only appears on a small number of training examples".

**Summary Of The Paper:**

Most existing papers working on spurious correlation focus on the correlation that existed in the majority of the examples. This paper discovered that even a few spurious examples can still lead to the model learning the spurious correlation. Moreover, these "rare" spurious correlations bring negative effects regardless of the strength of spurious patterns, network architectures, and optimization methods.  Finally, the paper provides a theoretic analysis of a simple binary classification model. Based on the analysis, a few techniques are proposed to improve learning with rare spurious correlations.

**Summary Of The Review:**

The paper discovers that even a few spurious examples can still lead to the model learning the spurious correlation. Many interesting results are shown with experiments: It shows that spurious patterns with larger empirical norms can cause spurious correlation more easily, and network architectures with higher sensitivity to its input are more susceptible to learning spurious correlations.

---

> ### Author Response · Authors · 2022-11-08
> **Response**
>
> - Does the technique proposed also work when the majority of the data present spurious correlation, e.g. Adding l2 regularization? Because as mentioned in the paper: l2 regularization only "suppresses the use of features that only appears on a small number of training examples".
>
> In the setting where spurious correlation is caused by having a significant portion of training examples (non-rare setting), L2 regularization is important for mitigating spurious correlation, as pointed out by [1]. However, the mechanism is different from our rare setting. In the non-rare setting, mitigating spurious correlation generally involves reweighting each sample based on the spurious attribute and the sample frequency. According to [2], the importance of regularization appears to be more of a necessary step for the generalization after the reweighting of each sample. This is different from suppressing features from rare examples.
>
> [1] Sagawa, Shiori, et al. "Distributionally robust neural networks for group shifts: On the importance of regularization for worst-case generalization." arXiv preprint arXiv:1911.08731 (2019).
>
> [2] Kulynych, Bogdan, et al. "What You See is What You Get: Distributional Generalization for Algorithm Design in Deep Learning." arXiv preprint arXiv:2204.03230 (2022).

---

### Official Review · Reviewer_LUVv · 2022-10-24

**Confidence:** 4
**Correctness:** 3
**Technical Novelty And Significance:** 2
**Empirical Novelty And Significance:** 2
**Recommendation:** 5

**Clarity, Quality, Novelty And Reproducibility:**

The paper contains a lot of content and is easy to follow which is good, but the main observation and conclusion are a bit weak for me.

**Strength And Weaknesses:**

Strength:
1. The paper studies an interesting topic of learning with rare spurious correlations via both experimental and theoretical approaches.
2. The paper studies whether rare spurious correlations are learned on both synthetic and real data.
3. The paper studies the consequences of rare spurious correlations from a privacy and test accuracy perspective.
4. The paper introduces simple and effective methods suggested by theoretical evidence to reduce the undesirable consequence of spurious correlation.
5. The paper is well written.

Weaknesses and questions:

1. In Figure 1, it seems like the seven different spurious patterns (adding different kinds of noise) are not strong enough to generate a spurious correlation. How about using more challenging datasets, such as Colored MNIST?

2. The main observations that neural networks can learn rare spurious correlations with few spurious examples are a little weak for me. It is not a big surprise to find out that adding perturbated patterns into training samples can impact the confidence of prediction on target classes during inference.


**Summary Of The Paper:**

This paper investigates how sensitive neural networks are with respect to rare spurious correlations from three perspectives. First, studying how many training points with the spurious pattern would cause noticeable spurious correlations in the synthetic and real datasets. Second, studying how rare spurious correlations affect neural networks’ privacy and test accuracy. Third, studying how to mitigate the effects of rare spurious correlation.


**Summary Of The Review:**

With the findings above, I currently give the paper a borderline score.

---

> ### Author Response · Authors · 2022-11-08
> **Response**
>
> - In Figure 1, it seems like the seven different spurious patterns (adding different kinds of noise) are not strong enough to generate a spurious correlation. How about using more challenging datasets, such as Colored MNIST?
>
> We would like to clarify that from the results in Figure 2, we have shown that the spurious correlations generated by the spurious patterns in Figure 1 are learnt by the neural network. Based on what we have observed in Appendix B2, where larger an L2 norm of the spurious pattern leads to more spurious correlations learnt, we could expect the models trained on Colored MNIST to learn more spurious correlation as changing the background color leads to a large change in terms of the L2 distance.
>
> - The main observations that neural networks can learn rare spurious correlations with few spurious examples are a little weak for me. It is not a big surprise to find out that adding perturbated patterns into training samples can impact the confidence of prediction on target classes during inference.
>
> We would like to point out that in addition to the observation that neural networks can learn rare spurious correlations with few spurious examples, we also have many other important findings that have not been revealed before.
> We have shown that rare spurious correlations could happen in natural scenes and cause privacy concerns. These two important aspects have not been unveiled before.
> Methods for completely mitigating rare spurious correlations are still unknown. We provided a detailed analysis of how it could be moderately mitigated, which serves as an important stepping stone for future work.

---

### Official Review · Reviewer_bwQt · 2022-10-25

**Confidence:** 3
**Correctness:** 3
**Technical Novelty And Significance:** 3
**Empirical Novelty And Significance:** 3
**Recommendation:** 5

**Clarity, Quality, Novelty And Reproducibility:**

Clarity is good. The content is well-organized and easy to follow. This work provides extensive experimental evaluations and sufficient details. Novelty is fair.

**Strength And Weaknesses:**

Strength:

1. The paper is well-motivated, organized, easy to follow and well-written.
2. The study aims to answer fundamental research questions, which can provide new insights to the community.
3. The authors provide theoretical justification for their findings

Weakness:
1. The technical contribution is somewhat limited in the sense that prior works have already shown that neural networks are more biased towards "easy-to-learn" spurious attributes.
2. A major chunk of the experimental analysis is based on cases where the spurious correlation is injected synthetically. It would be more interesting if the authors also showed results on commonly studied spuriously correlated datasets such as Waterbirds, and CelebA.

**Summary Of The Paper:**

This paper is dedicated to the problem of spurious correlations. The work systematically studies fundamental questions such as how many spuriously correlated training points are necessary for a neural net to get biased towards learning it. Specifically, it investigates the domain in which spurious correlations are rare. Interestingly the study indicates that even a single spuriously correlated sample can bias the learning of a neural network. Finally, the authors highlight three regularization methods for mitigating spurious correlations.

**Summary Of The Review:**

The paper studies some important research questions in the domain of spurious correlations. The findings also provide non-trivial contributions to the community. However, because of the synthetic setup, the overall impression of this work is borderline.  I suggest the authors comment on the weaknesses mentioned above and I am open to changing my rating.

---

> ### Author Response · Authors · 2022-11-08
> **Response**
>
> - The technical contribution is somewhat limited in the sense that prior works have already shown that neural networks are more biased towards "easy-to-learn" spurious attributes.
>
> We agree that prior works in shortcut learning and backdoor attack have already shown that neural networks can be biased towards "easy-to-learn" spurious attributes (shortcut features). However, all these works did not focus on the situation where the appearance of spurious examples is extremely rare, e.g., even just one spurious example. This is extremely crucial for the identification of the privacy concern as privacy could be less of a concern if many examples with the same feature are required to cause neural networks to learn the spurious correlations.
>
> - A major chunk of the experimental analysis is based on cases where the spurious correlation is injected synthetically. It would be more interesting if the authors also showed results on commonly studied spuriously correlated datasets such as Waterbirds, and CelebA.
>
> In Waterbirds and CelebA, the spurious correlations are caused by having a significant portion of spurious examples in the training set, which is different from our setting. There are major differences between rare and non-rare settings as many of the techniques in the non-rare setting, such as reweighting and detection, will not work in the rare setting.
>
> As prior techniques are unusable in the rare setting, we focus on the analysis with synthetic injection. If we are not even able to completely mitigate rare spurious correlations in the synthetic injection case (which is still true in the current research field), there is no hope of completely mitigating spurious correlations in the more general case.

---

### Author Response · Authors · 2022-11-08
**General Comments**

We thank all the reviewers for their insightful and constructive feedback. To clarify, the key messages of this work are to:
1. Illustrate that just one spurious example could cause the network to learn spurious correlations with either natural or artificial patterns. Although there are prior works suggesting few examples could cause the model to learn some spurious pattern, they have not studied this low number of spurious examples.
2. Show that as extremely rare spurious examples cause the rare spurious correlations, they can lead to privacy risks and misclassification when these spurious features are present.
3. Provide a method backed by theoretical and empirical results for mitigating rare spurious correlations.

We summarize two common concerns among the reviewers and provide clarifications and responses. For individual questions, we will reply separately to avoid complications.

The first common concern (raised by Reviewer bwQt and yphb) is about the use of synthetic patterns for spurious patterns. We emphasize that how neural networks pick up rare spurious correlations is still an active research area. The mechanism is still largely unknown. Under this circumstance, it would be better to start with synthetic patterns to gain a solid understanding. For instance, the example in section 5 allows us to observe the phase transition between having and not having spurious examples. It also gives us insights into how some regularization methods can help with mitigating rare spurious correlations.

The second common concern (raised by Reviewer LUVv and yphb) is regarding the novelty of discovering the rare spurious correlation. We want to point out that prior work, such as shortcut learning and backdoor triggers, did not operate in the regime of 1 to 10 spurious examples (they usually operate at 50+ spurious examples). This makes a significant difference in the privacy concern as being picked out as the only spurious example is significantly worse than being picked out as one of 50 spurious examples. Thus, this allows us to have more fruitful results other than discovering the rare spurious correlation itself. We have more insights on privacy consequences and the mitigation of rare spurious correlations, which are new.

---

### Author Response · Authors · 2022-11-23
**Any additional feedback or questions?**

We thank all the reviewers for their thoughtful reviews and comments. With the discussion period towards the end, we would like to see if there are any additional comments or questions we can help address. We are happy to answer any further questions.

Thanks!

---

### Decision · Program_Chairs · 2023-01-20

**Decision:**

Reject

**Justification For Why Not Higher Score:**

See the above explanations of weaknesses of the submission

**Justification For Why Not Lower Score:**

I recommend rejection.

**Metareview: Summary, Strengths And Weaknesses:**

The paper studies  the problem of spurious correlations specially when they are rare. In a setting, they found that even a single spuriously correlated sample can bias the learning of a neural network. The paper is well-motivated, organized, easy to follow and well-written. Reviewers generally liked the fundamental research questions asked in the paper. However, there are some critical weaknesses in the paper : Although the paper provides some interesting analysis, most results are based on a synthetic setup. I believe that the paper could have been improved with more results and empirical analysis on natural benchmarks. Connections between the work and backdoor triggers should be explained more clearly. Given all, I think the paper needs a bit more work before being accepted.